# Specific Binding Ratio Estimation of [^123^I]-FP-CIT SPECT Using Frontal Projection Image and Machine Learning

**DOI:** 10.3390/diagnostics13081371

**Published:** 2023-04-07

**Authors:** Akinobu Kita, Hidehiko Okazawa, Katsuya Sugimoto, Nobuyuki Kosaka, Eiji Kidoya, Tetsuya Tsujikawa

**Affiliations:** 1Radiological Center, University of Fukui Hospital, 23-3, Matsuoka-Shimoaizuki, Eiheiji-cho, Fukui 910-1193, Japan; 2Biomedical Imaging Research Center, University of Fukui, 23-3, Matsuoka-Shimaizuki, Eiheiji-cho, Fukui 910-1193, Japan; 3Department of Radiology, Faculty of Medical Sciences, University of Fukui, 23-3, Matsuoka-Shimoaizuki, Eiheiji-cho, Fukui 910-1193, Japan

**Keywords:** dopamine transporter (DAT), single-photon emission-computed tomography (SPECT), imaging, deep learning (DL), convolutional neural network, specific binding ratio (SBR)

## Abstract

This study aimed to develop a new convolutional neural network (CNN) method for estimating the specific binding ratio (SBR) from only frontal projection images in single-photon emission-computed tomography using [^123^I]ioflupane. We created five datasets to train two CNNs, LeNet and AlexNet: (1) 128FOV used a 0° projection image without preprocessing, (2) 40FOV used 0° projection images cropped to 40 × 40 pixels centered on the striatum, (3) 40FOV training data doubled by data augmentation (40FOV_DA, left-right reversal only), (4) 40FOVhalf, and (5) 40FOV_DAhalf, split into left and right (20 × 40) images of 40FOV and 40FOV_DA to separately evaluate the left and right SBR. The accuracy of the SBR estimation was assessed using the mean absolute error, root mean squared error, correlation coefficient, and slope. The 128FOV dataset had significantly larger absolute errors compared to all other datasets (*p* < 0. 05). The best correlation coefficient between the SBRs using SPECT images and those estimated from frontal projection images alone was 0.87. Clinical use of the new CNN method in this study was feasible for estimating the SBR with a small error rate using only the frontal projection images collected in a short time.

## 1. Introduction

Single-photon emission-computed tomography (SPECT) can show the distribution of the dopamine transporter (DAT) in the brain using [^123^I] ioflupane ([^123^I]-FP-CIT), the DAT imaging ligand. Employing [^123^I]-FP-CIT SPECT increases the precision of choosing the best treatment options and aids in the early detection of Parkinsonian syndrome and dementia with Lewy bodies [1,2,3]. Visual evaluation is reportedly sufficient for diagnosis [1,4]; however, diagnostic performance is enhanced by quantitative evaluation in addition to visual inspection [5,6,7]. The specific binding ratio (SBR), calculated as the ratio between specific and non-specific accumulation, is the standard quantification parameter in [^123^I]-FP-CIT SPECT. Currently, in clinical practice, quantitative evaluation by SBR plays an important role as an adjunct to visual evaluation.

Worldwide, [^123^I]-FP-CIT SPECT exams are often conducted, and clinical practice guidelines have been published in various regions [8,9]. [^123^I]-FP-CIT SPECT scans in Japan are performed in accordance with the ‘Clinical Ioflupane Guidelines’ of the Japanese Society of Nuclear Medicine, available online at http://www.jsnm.org/archives/1151/ (accessed on 6 April 2023). To get sufficient acquisition counts, these guidelines indicate an appropriate acquisition time of 30 to 45 min. Short-time scans reduce the acquisition counts, deteriorate the quality of the visual evaluation, and underestimate SBR [10]. In addition, a multicenter study conducted in Japan revealed that all 14 participating facilities used an acquisition time of approximately 30 min [11], suggesting that many hospitals in Japan comply with the guidelines. However, patients with claustrophobia, back pain, dementia, etc., often request shorter scans or want to stop immediately after the scan is started. If the examination is terminated without sufficient scanning time, a tomographic image may not be obtained, and the SPECT study would fail to obtain SBR and any other information. Although [^123^I]-FP-CIT SPECT is important to evaluate striatal functions, some patients cannot tolerate the static position for reasons described above. If a frontal projection image with only a short time duration can estimate striatal functions in such cases, the image analysis method may provide appropriate clinical diagnosis.

Our study aimed to develop a new CNN method for estimating SBR from only frontal projection images in [^123^I]-FP-CIT SPECT, and it also aimed to evaluate the error and correlation between SBR estimated by the new CNN method and SBR obtained by the full SPECT study. If patients who cannot undergo the exam for the required amount of time could receive an accurate estimate of SBR from a rapid frontal projection, the results of our investigation would be of great clinical value. Nakajima et al. developed a machine learning (ML) method that could diagnose Parkinsonian syndrome and dementia with Lewy bodies [12]. They obtained more accurate results than conventional ROI-based methods. However, the purpose of their study was to improve diagnostic accuracy using commonly obtained SPECT images, and the concept was different from our study.

## 2. Materials and Methods

### 2.1. SPECT Imaging

Four hundred twenty patients (72.1 ± 9.7 y) who underwent [^123^I]-FP-CIT SPECT at the University of Fukui Hospital from January 2017 to December 2021 were included in this retrospective study. The Ethics Committee of our university’s Faculty of Medical Sciences approved the study’s protocol (#20210156) and waived the requirement of obtaining patient informed consent and approved a retrospective evaluation of imaging and clinical data. Patients received a gradual intravenous infusion of approximately 170 MBq [^123^I]-FP-CIT in the morning, and 3–4 h later, they had SPECT scans (167 MBq at noon).

A dual-head gamma camera (Symbia T2, Siemens, Erlargen, Germany) and low- to medium-energy general-purpose collimators were installed in the SPECT/CT scanner. The SPECT images were taken with a 1.45 zoom, a 3.3 mm pixel size, and a 10% scan energy window in close proximity to the 159-keV photo-peak. With the triple-energy window scatter correction (SC), two sub-windows at 7% were constructed on the top and bottom sides of the photo-peak window [13]. With an angular range of 4° increments, 90 frames were used to get the SPECT data. The camera heads were as near to the patient’s head as possible. An attenuation correction low-dose CT scan was performed following the gathering of SPECT data. The data were acquired in continuous rotation mode at 14 min × two repeats. With eight iterations and ten subsets, the SPECT data were reconstructed using an iterative three-dimensional-ordered subset expectation maximization algorithm. After 1.5× expansion, the pixel size of each slice in a 128 × 128 matrix was 2.2 mm, and all images were subjected to a Gaussian filter with a full width at half maximum of 6 mm. Two image sets were reconstructed from the image data with the correction of CT attenuation and SC (ACSC) and without any correction (non-AC).

### 2.2. SBR Calculation

The SBR values were calculated with DaT View (AZE, Tokyo, Japan) using the Tossici-Bolt and count-based automatic calculation methods written as a Matlab script [14,15,16]. DaT View is a widely used SBR calculation software in Japan that recommends using non-AC images as input images [17,18]. In the count-based method (the most commonly used calculation method in the world), SBR is calculated from region of interest (ROI) counts based on the formula (striatum ROI—BG)/BG, where BG is background counts. Our semi-automatic calculation method used in this study can be applied for either ACSC or CTAC images, and ACSC is recommended [15,16]. Thus, we applied non-AC images for DaT View, and we applied ACSC images for the count-based method.

### 2.3. Network Architecture

Only the front (0°) projection image from the projection image acquired by SPECT was defined as the 0° projection image. The new CNN method for SBR estimation was constructed using CNN upon studying the combination of the 0° projection image and the SBRs from the full SPECT data.

The 0° projection image was a total image of 4 degrees (0° to 3°) of the angle at the time of SPECT continuous rotation acquisition, which was equivalent to 37.3 s of acquisition time. Two types of datasets were constructed: one for estimating the left and right average SBR values, and the other for estimating the left and right SBRs independently. The Neural Network (v2. 10, Sony Network Communications Inc., Tokyo, Japan) was used for deep learning (DL) in our CNN models for SBR estimation from the 0° projection image. Input (I), rectified linear unit (R, ReLU), convolution (C), max-pooling (M), affine (A), Tanh (T), and squared error (S) made up the architecture used in this investigation (Appendix A). The layer’s function is indicated by square boxes, and the specifications for each layer are depicted in the figures on the right side of the box. The three values on the right side of the first input layer, for example, denote the number of colors (“1” signifies monochrome) and the size of the input image (height and width). In the second convolution layer, the output feature map’s number and size (height and width) were both displayed in the same format. The rectified linear unit is referred to as “ReLU”. The home page describes additional details in this figure (Sony Network Communications Inc., 2020, available online at https://dl.sony.com/ (accessed on 6 April 2023)). We tested two types of networks, LeNet [19] (Figure 1a) and AlexNet [20] (Figure 1b), based on CNN.

CNN is one of the commonly used Deep Learning architecture types for identifying and classifying images. It has several layers, including fully connected, convolutional, and pooling layers. The convolutional layer uses filters to extract features from the input image, the pooling layer minimizes processing by downsampling the image, and the fully connected layer provides the final prediction. Innovative CNN topologies such as LeNet and AlexNet were developed to increase accuracy and lower computing costs. LeNet, also known as the classic neural network, was developed by LeCun et al. in the 1990s for the recognition of handwritten and machine-printed characters [19]. The architecture is relatively uncomplicated and easy to comprehend. However, AlexNet has more layers and more filters than LeNet, making the network more complex and more accurate.

### 2.4. Datasets

Five datasets were created. Figure 2 displays the images utilized in each dataset and the quantity of training, validation, and test cases. The 128FOV dataset consisted of a combination of 260, 80, and 80 cases of training, validation, and test data, respectively (Figure 2a), and used a 0° projection image without preprocessing. Table 1 shows the characteristics of the patients included in the datasets. No significant differences were observed with respect to their age and gender. They were suspected to have Parkinsonian syndrome or dementia with Lewy bodies. The dataset 40FOV used 0° projection images cropped to 40 × 40 pixels centered on the striatum (the number of testing, validation, and training cases was the same as for 128FOV) (Figure 2b). The dataset 40FOV_DA was the 40FOV training data doubled by data augmentation (left-right reversal only), with 520, 80, and 80 cases of training, validation, and test data, respectively (Figure 2c). In addition, for the 40FOV and 40FOV_DA, the 40 × 40 pixels cropped image was split into left and right (20 × 40) so that the left and right SBR could be evaluated separately. We named those datasets 40FOVhalf and 40FOV_DAhalf, respectively (Figure 2d,e).

### 2.5. Evolution

The SBR obtained by DaT View and the count-based method were defined as SBR_TB_ and SBR_cb_, respectively. The SBR estimated by DL using 0° projection images was defined as SBR_DL_. Using the root mean square error (RMSE), correlation coefficient (Corr) mean absolute error (MAE), and slope, we assessed the accuracy of the SBR estimate. In addition, the CNN-focused regions of the 0° projection image were identified by Gradient-weighted Class Activation Mapping (Grad-CAM) [21]. According to their significance, Grad-CAM created a heat map of the pixels that CNN concentrated on.

### 2.6. Statistical Analysis

To compare absolute errors between datasets, an analysis of variance (ANOVA) was performed, followed by Tukey’s test for post hoc testing. To compare the MAE, RSMU, and Corr across the AlexNet and LeNet networks, the Wilcoxon signed-rank test was utilized. EZR ver. 1.41 [22] was used for the statistical analysis, and *p* < 0.05 was regarded as significant.

## 3. Results

Figure 3 shows the scatter plots of the conventional SBR and the SBR for each dataset and network, and the corresponding Bland–Altman plot for each SBR. SBRBT vs. SBRDL is shown in Figure 3a, and SBRcb vs. SBRDL is shown in Figure 3b. The solid line is the line of identity. The Bland–Altman plot shows that 128FOV has a distribution with a biased plot compared to 40FOV and 40FOV_DA. The trend was similar for both the DaT View and the count-based methods. Table 2 shows the values of the MAE, RSMU, Corr, and slope obtained from the DaT View and count-based method. The 128FOV had larger MAEs and RMSEs and smaller Corrs and slopes compared to the other datasets. The 128FOV had significantly larger MAEs compared to the other datasets (*p* < 0.05). Apart from the 128FOV combination, there was no significant variation in MAE and RMSE, and the values did not differ considerably between AlexNet and LeNet. Data augmentation had slightly improved estimates. SBR_BT_ and SBR_cb_ showed similar trends.

Figure 4 shows the original 0° projection image and the heat map image of Grad-CAM for 128FOV (Figure 4a, top) and 40FOV (Figure 4a, bottom). In the Grad-CAM of 128FOV, the many background dots indicate that the CNN focused on the background and the brain in many cases. The 40FOV of the same case at the bottom shows that the small FOV was centered on the striatum, and the CNN focused on the brain. Figure 4b includes [^123^I]-FP-CIT uptake in the parotid gland. In the Grad-CAM of the 128FOV (Figure 4b, top), the color dots in the parotid gland indicate that the CNN focused on the parotid gland in addition to the background. On the other hand, the image of the 40FOV (Figure 4b, bottom) was cropped around the striatum, and the parotid gland was out of the field of view, indicating that the CNN was not focused on parotid accumulation.

## 4. Discussion

The purpose of this study was to create a new CNN method for estimating SBR from only frontal projection images in [^123^I]-FP-CIT SPECT using CNN and to evaluate the error and correlation between the SBR estimated by the new CNN method and the full SPECT study. The new CNN method estimated the SBR with high correlation and low error using the preprocessed dataset for the projection image. In addition, although the dataset for evaluating the left and right SBR separately (40FOVhalf and 40FOV_DAhalf) had larger errors than the 40FOV and 40FOV_DA, it demonstrated a strong correlation between the SBRs with full SPECT data and the SBR_DL_. Bland–Altman plots revealed that the 128FOV dataset had a larger systematic error than the 40FOV and 40FOV_DA datasets. This suggests that the increase in cases did not improve the SBR estimation accuracy in the 128FOV dataset. Therefore, 128FOV projection data alone (image without trimming processing) are inappropriate for SBR estimation. It seems that noise in the large space outside the head may have affected this result because 40FOV shows a slightly similar trend. These results showed the same tendency in both the DaT View and the count-based methods. We consider calculating the left and right SBR independently to be clinically valuable since patients with Parkinsonian syndrome often have left–right differences in SBR [23]. Furthermore, the number of data is doubled by dividing the data into left and right, which improves the accuracy of DL. Since this study yielded low error rates, the results may not be equivalent to full SPECT data (Figure 3). Therefore, our method may be used as an alternative to SBR only when collected for a short scanning time.

The results using the 128FOV were significantly poorer than the other datasets. The first possible reason for this could be that the CNN often focused equally on the background and on the brain (Figure 4a). This was probably because the area occupied by the background was larger than the area occupied by the brain in the images of the 128FOV. Although the background signal was not involved in the SBR calculation, the background noise was assumed to induce confusion in the learning process. Furthermore, in cases with accumulation in the salivary glands (Figure 4b), the accumulation could affect the learning process with 128FOV. Physiological accumulation of the salivary glands should be excluded from the calculation of SBR [24]. Therefore, the projection data without trimming are considered inappropriate for SBR estimation. The results of the estimation using data augmentation approaches were marginally better than those without data augmentation techniques, indicating that the accuracy of the SBR estimation was enhanced by the increase in the number of cases. The accuracy of the SBR estimation would be enhanced by additional data augmentation, which would increase the number of cases and training data.

There was no significant difference between the two networks. Therefore, we believe that by optimizing these networks and testing networks with more layers, such as VGGNet [25] and ResNet [26], we could find the optimal network for SBR estimation; moreover, further improvement can be anticipated in terms of accurate SBR estimation.

## 5. Conclusions

The 0° projection image in [^123^I]-FP-CIT SPECT estimated the SBR with high correlation and low errors by using a preprocessed dataset for the projection image. The estimation accuracy was improved by increasing the training cases through data augmentation. Clinical use of the new CNN method in this study is considered feasible since SBR can be estimated with a small error rate using only frontal projection images collected in a short time.

## Figures and Tables

**Figure 1 diagnostics-13-01371-f001:**
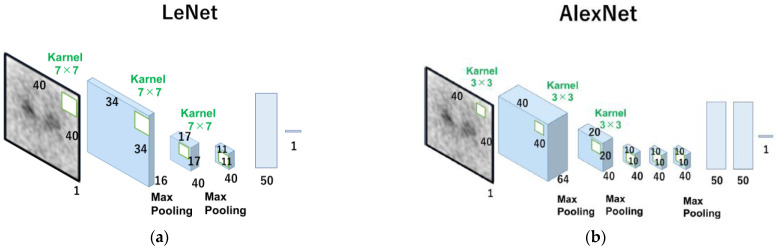
Scheme of the algorithms of ReNet and Alexnet used in this study. The number under the square boxes is the number of filters. LeNet (**a**), the basic shapes of CNN, consists of alternating layers of convolutional and pooling layers, and AlexNet (**b**) consists of five convolutional layers and three fully connected layers.

**Figure 2 diagnostics-13-01371-f002:**
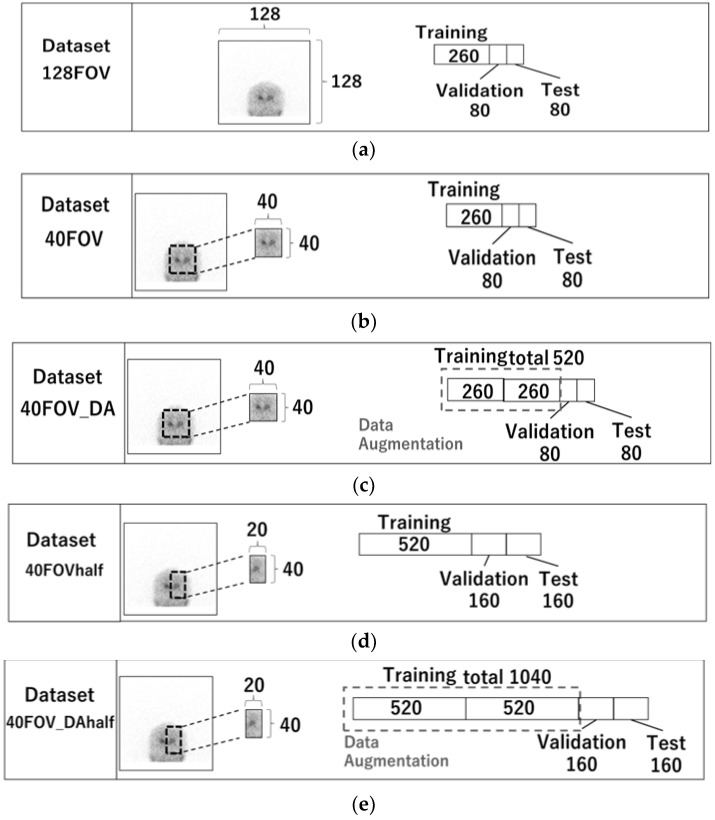
128FOV: combination of 260, 80, and 80 cases of training, validation, and test data, respectively. (**a**) 128FOV used a 0° projection image without preprocessing. (**b**) 40FOV used 0° projection images cropped to 40 × 40 pixels centered on the striatum (the number of testing, validation, and training cases was the same as for 128FOV). (**c**) 40FOV_DA used 40FOV training data doubled by data augmentation (left-right reversal only), with 520, 80, and 80 cases of training, validation, and test data, respectively. (**d**) 40FOVhalf used the image from 40FOV, which was divided in half (20 × 40) at the center of the left and right striatum, with 520, 80, and 80 cases of training, validation, and test data, respectively. (**e**) 40FOV_DAhalf used the image from 40FOV_DA, which was divided in half (20 × 40) at the center of the left and right striatum, with 1040, 80, and 80 cases of training, validation, and test data, respectively.

**Figure 3 diagnostics-13-01371-f003:**
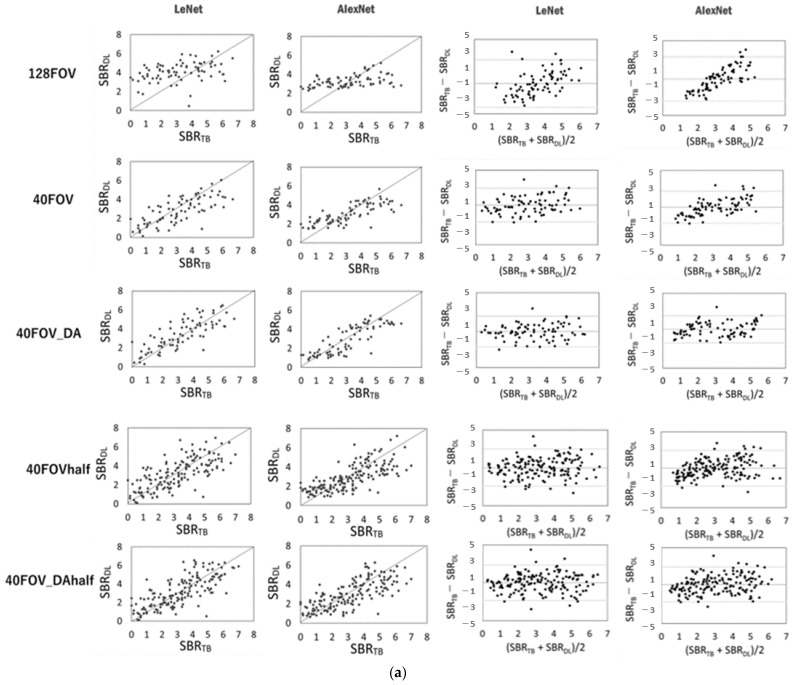
Scatter plots of conventional specific binding rate (SBR) (SBR_BT_ or SBR_cb_) and SBR (SBR_DL_) for each dataset and network using the DaT view (**a**, left half) and the count-based methods (**b**, left half). The solid line is the line of identity. Right half shows the corresponding Bland–Altman plot for each SBR. Black lines represent the mean bias; the upper and lower dotted lines are 1.96 standard deviation.

**Figure 4 diagnostics-13-01371-f004:**
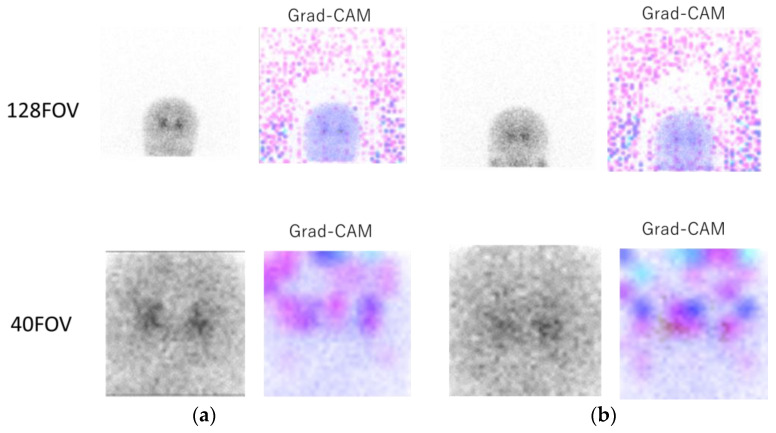
Upper part, image of the 128FOV and Gradient-weighted Class Activation Mapping (Grad-CAM)’s heat map; lower part, image of the 40FOV of the same case and the heat map image of Grad-CAM. (**a**) In the Grad-CAM of 128FOV, the many background dots indicate that the convolutional neural network (CNN) focused on the background. The 40FOV of the same case in the bottom shows that the small FOV was centered on the striatum, and CNN focused on the brain. (**b**) A case of [^123^I] ioflupane ([^123^I]-FP-CIT) accumulation in the parotid gland. In the Grad-CAM of the 128FOV (**top**), the color dots in the parotid gland indicate that the CNN focused on the parotid gland in addition to the background, while the image of the 40FOV (**bottom**) was cropped around the striatum, and the parotid gland was outside the field of view, indicating that the CNN was not interested in parotid accumulation.

**Table 1 diagnostics-13-01371-t001:** Patient characteristics.

	Training	Validation	Test
Number of cases	260	80	80
Age (y, mean ± SD)	72.2 ± 9.6	71.8 ± 10.3	72.4 ± 9.8
Men/women (N)	137/123	49/31	44/36

**Table 2 diagnostics-13-01371-t002:** Results of DaT View and count-based methods analysis. MAE, RSMU, Corr, and slope values for the datasets 128FOV, 40FOV, 40FOV_DA, 40FOVhalf, and 40FOV_DAhalf.

		LeNet				AlexNet			
	Dataset	MAE	RMSE	Corr	Slope	MAE	RMSE	Corr	Slope
DaT View	128FOV	1.56	1.87	0.36	0.21	1.21	1.51	0.44	0.16
	40FOV	0.92	1.18	0.74	0.60	0.86	1.09	0.78	0.47
	40FOV_DA	0.83	1.05	0.80	0.74	0.76	0.94	0.84	0.74
	40FOVhalf	1.00	1.25	0.71	0.67	0.98	1.21	0.72	0 60
	40FOV_DAhalf	0.94	1.20	0.74	0.73	0.95	1.18	0.73	0.61
Count-based methods	128FOV	0.58	0.72	0.41	0.23	0.63	0.76	0.25	0.10
	40FOV	0.35	0.42	0.84	0.70	0.48	0.57	0.84	0.36
	40FOV_DA	0.35	0.44	0.83	0.76	0.35	0.42	0.87	0.59
	40FOVhalf	0.45	0.55	0.73	0.59	0.39	0.50	0.78	0.64
	40FOV_DAhalf	0.40	0.49	0.80	0.71	0.40	0.50	0.77	0.62

MAE: mean absolute error, RMSE: root mean squared error, Corr: correlation coefficient. 128FOV had a larger MAE and RMSE than the other datasets, and the Corr and slope was small; only weak correlations were obtained. When AlexNet was used for the network, there was little difference in the values when LeNet was used. Data augmentation had a slight effect, slightly improving estimates. SBR_BT_ and SBR_cb_ showed similar trends.

## Data Availability

The datasets used and/or analyzed during the current study are available from the corresponding author upon reasonable request.

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
