# Peer review of "Specific Binding Ratio Estimation of [123I]-FP-CIT SPECT Using Frontal Projection Image and Machine Learning"

_diagnostics, 2023, doi:10.3390/diagnostics13081371_

Round 1
Reviewer 1 Report
See separate file
This manuscript describes a novel method that aims to estimate [123
I]-FP-CIT specific binding ratios using only (short) frontal projection images together with machine learning. The method could be of interest to the field, but requires attention to a number of issues:
1. The methodology should be described in more detail. Some suggestions:
a. SBR was calculated using DaT View applied to non-AC images, but also using a count-based method applied to ACSC images. However, details about the count-based method are lacking. In addition, the authors should specify why these two methods were considered in parallel.
b. According to the methods both low- and medium-energy general-purpose collimators were used. This is strange. Were these collimators used for different patients? If so, it seems that data cannot be combined.
c. The triple-energy window scatter correction should be described or, at least, a reference should be given.
d. The underlying principles of both neural networks should be briefly described (readers of the journal may not be familiar with neural networks). What is the difference between both neural networks? Note that Figure 1 is unclear, i.e. the text in Figure 1 is unreadable.
e. The patient characteristics of training, validation and test data sets should be provided and the authors should check that there are no significant differences between data sets,
2. 260, 80, and 80 cases of training, validation, and test data, respectively, were used to assess the methods. However, this is a total of 420 cases, which is higher than the 400 subjects included in the entire study. Where do the additional 20 cases come from?
3. In addition, it is questionable whether the use of 520 training cases is a valid approach, as in this case left and right data are derived from the same patient as independent data, which clearly they are not. The validity of this approach should be discussed.
4. The most important issue is the fact that all correlation plots show a substantial level of scatter. To illustrate individual variability, the use of correlation plots is not sufficient. The authors should also present Bland-Altman plots.
5. In the presence of variability, correlation may still be good in which case the method could be used for group analyses, but diagnosing individual patients is quite another matter. To address this question, it would be useful to see how diagnosis of individual patients would change (or not) when the new method is used.
Author Response
Response to Reviewer 1
Thank you so much for your kind and thoughtful comments, and giving us the opportunity to revise our manuscript. We have considered carefully every point raised and revised the manuscript based on your comments and suggestions. Revisions are indicated by red texts. We hope this manuscript is now successfully revised according to your kind suggestions.
1. The methodology should be described in more detail. Some suggestions:
- SBR was calculated using DaT View applied to non-AC images, but also using a count-based method applied to ACSC images. However, details about the count-based method are lacking. In addition, the authors should specify why these two methods were considered in parallel.
-
Answer:
According to your thoughtful comment, we have added the following brief explanation of the SBR calculation methods (Line 97-103):
DaT View is a widely used SBR calculation software in Japan that recommends using non-AC images as input images [17,18]. In the count-based method, the most commonly used calculation method in the world, SBR is calculated from region of interest (ROI) counts based on the formula of (striatum ROI – BG)/ BG, where BG is background counts. Our semi-automatic calculation method used in this study can be applied for either ACSC or CTAC images, and ACSC is recommended [15,16]. Thus, we applied non-AC images for DaT View and ACSC images for the count-based method in SBR calculation.
-
- According to the methods both low-and medium-energy general-purpose collimators were used. This is strange. Were these collimators used for different patients? If so, it seems that data cannot be combined.
-
Answer:
We are sorry about this mistake.
The correct description of 'low- and medium-energy general-purpose collimators ' is 'low- to medium-energy general-purpose collimators ', and only LMEGP collimators were used in this study.
We have corrected that part in the manuscript. (Line 79)
-
- The triple-energy window scatter correction should be described or, at least, a reference should be given.
-
Answer:
A Reference [13] have been added according to the reviewer’s suggestion. (Line 83)
-
- The underlying principles of both neural networks should be briefly described (readers of the journal may not be familiar with neural networks). What is the difference between both neural networks? Note that Figure 1 is unclear, i.e. the text in Figure 1 is unreadable.
-
Answer:
We would like to this thoughtful comment. We added the following brief explanation of the basic principles of neural networks (Line 127-136). And according to the reviewer’s suggestion, the algorithms of ReNet and Alexnet used in this study have been added in Figure 1 and original figures of Fig. 1 was moved to a Supplement Figure 1 at the last page.
CNN is a one of the commonly used Deep Learning architecture types for identifying and classifying images. It has several layers, including fully connected, convolutional, and pooling layers. The convolutional layer extracts features from the input image using filters, the pooling layer reduces computation by downsampling the image, and the fully connected layer makes the final prediction. Innovative CNN topologies like LeNet and AlexNet were developed to increase accuracy and lower computing costs. LeNet, also known as the classic neural network, was created by LeCun et a. in 1990s for the recognition of handwritten and machine-printed characters [19]. The architecture is relatively uncomplicated and easy to comprehend. However, AlexNet has more layers and more filters than LeNet, making the network more complex and more accurate.
-
- The patient characteristics of training, validation and test data sets should be provided and the authors should check that there are no significant differences between data sets,
-
Answer:
We have added a table as Table 1 and the following description according to the reviewer’s suggestion. (Line149-152)
Table 1 shows the characteristics of the patients included in the datasets. No significant differences were observed with respect to their age and gender. They were suspected to be Parkinsonian syndrome or dementia with Lewy bodies.
-
2. 260, 80, and 80 cases of training, validation, and test data, respectively, were used to assess the methods. However, this is a total of 420 cases, which is higher than the 400 subjects included in the entire study. Where do the additional 20 cases come from?
Answer:
We apologize for this mistake. The correct description for the total number of subjects in the study is 420. We have corrected the number. (Line 70)
3. In addition, it is questionable whether the use of 520 training cases is a valid approach, as in this case left and right data are derived from the same patient as independent data, which clearly they are not. The validity of this approach should be discussed.
Answer:
We are sorry about lacking in clear explanation. In clinical practice, it is common to observe laterality by calculating and evaluating the SBR of the left and right striatum separately. Furthermore, the number of data is doubled by dividing the data into left and right, which improves the accuracy of DL. Therefore, we think this approach is appropriate. We have added this explanation in the first paragraph of Discussion. (Line274-275)
4. The most important issue is the fact that all correlation plots show a substantial level of scatter. To illustrate individual variability, the use of correlation plots is not sufficient. The authors should also present Bland-Altman plots.
Answer:
We agree with you and have added Bland-Altman plots in Fig.3. (see page 7-8) We have modified the first paragraph of the Results accordingly, and added the following discussion in page 10. (Line 267-271)
Bland–Altman plots revealed that the 128FOV dataset had a larger systematic error than the 40FOV and 40FOV_DA datasets. This suggests that increase in cases did not improve the SBR estimation accuracy in the 128FOV dataset. Therefore, 128FOV projection data alone (image without trimming processing) are inappropriate for SBR estimation.
5. In the presence of variability, correlation may still be good in which case the method could be used for group analyses, but diagnosing individual patients is quite another matter. To address this question, it would be useful to see how diagnosis of individual patients would change (or not) when the new method is used.
Answer:
We understand the reviewer’s concern, however, the objective of this study was to assess the error by attempting to obtain SBR from frontal projection alone in a short time for cases who were unable to complete examination with sufficient time. In a future study, we would like to evaluate accuracy of individual diagnosis when the new method is used.
Reviewer 2 Report
1. Add some empirical results in the abstract section and compare them with the existing ones.
2. Please write the motivation of your study in the introduction before your contributions.
3. Please summarize some relevant and latest survey articles and compare your contributions with the existing surveys.
4. Describe clearly the simulation part.
5. Plagiarism is more than 20%, Please make it less than 10%.
6. Include the design and formulations of the algorithms of the proposed work.
7. Explain all the figures & graphs in research style in the revised version of the manuscript text.
8. Qualities of all the figures are poor. Improve the quality for the reader's attention.
Author Response
Response to Reviewer 2
Thank you so much for your kind and thoughtful comments, and giving us the opportunity to revise our manuscript. We have considered carefully every point raised and revised the manuscript based on your comments and suggestions. Revisions are indicated by red texts. We hope this manuscript is now successfully revised according to your kind suggestions.
- Add some empirical results in the abstract section and compare them with the existing ones.
Answer:
Since this study was not designed for parallel comparison, the best result obtained in the study has been added in the abstract as follows. (Line23-24)
The best correlation coefficient between the SBRs using SPECT images and those estimated from frontal projection images alone was 0.87.
- Please write the motivation of your study in the introduction before your contributions.
Answer:
According to the reviewer’s suggestion, the following sentences have been added to the Introduction. (Line53-57)
Although [123I]FP-CIT SPECT is important to evaluate of striatal functions, some patients cannot tolerate in static posision for reasons described above. If a frontal projection image with only a short time can estimate striatal functions in such cases, the image analysis method may provide appropriate clinical diagnosis.
- Please summarize some relevant and latest survey articles and compare your contributions with the existing surveys.
Answer:
Thank you for the important suggestion. The following description was added in the Introduction. (Line63-67)
Nakajima et al. developed a machine learning (ML) method that could diagnose Parkinsonian syndrome and dementia with Lewy bodies [12]. They obtained more accurate results than conventional ROI-based methods. However, the purpose of their study was to improve diagnostic accuracy using commonly obtained SPECT images, and the concept was different from our study.
- Describe clearly the simulation part.
Answer:
We have added the following explanation in the section of ‘2.3 Network architecture’. (Line127-136)
CNN is a one of the commonly used Deep Learning architecture types for identifying and classifying images. It has several layers, including fully connected, convolutional, and pooling layers. The convolutional layer extracts features from the input image using filters, the pooling layer reduces computation by downsampling the image, and the fully connected layer makes the final prediction. Innovative CNN topologies like LeNet and AlexNet were developed to increase accuracy and lower computing costs. LeNet, also known as the classic neural network, was created by LeCun et a. in 1990s for the recognition of handwritten and machine-printed characters [19]. The architecture is relatively uncomplicated and easy to comprehend. However, AlexNet has more layers and more filters than LeNet, making the network more complex and more accurate.
- Plagiarism is more than 20%, Please make it less than 10%.
Answer:
Thank you for pointing it out. We have tried to minimize plagiarism, but could you confirm it again?
- Include the design and formulations of the algorithms of the proposed work.
Answer:
According to the reviewer’s suggestion, the algorithms of ReNet and Alexnet used in this study have been added in Figure 1 and original figures of Fig. 1 was moved to a Supplement Figure 1 at the last page.
- Explain all the figures & graphs in research style in the revised version of the manuscript text.
Answer:
We apologize for the inconvenience. We have enlarged some figures and revised legends.
- Qualities of all the figures are poor. Improve the quality for the reader's attention.
Answer:
We apologize for the inconvenience. Figures have been enlarged and improved in quality.
Round 2
Reviewer 1 Report
Thank you for your response and the changes made to the manuscript. I have no further queries, except that it would be useful to include a paragraph in the discussion in which you explain/hypothesize why the Bland-Altman plots are not horizontal (i.e. why bias is dependent on the actual value).
Author Response
Response to Reviewer 1
I have no further queries, except that it would be useful to include a paragraph in the discussion in which you explain/hypothesize why the Bland-Altman plots are not horizontal (i.e. why bias is dependent on the actual value).
Answer:
Thank you so much for your kind and thoughtful review.
We have added the following discussion (Line 270-271).
It seems that noise in the large space outside the head may have affected this result because 40FOV shows a slightly similar trend.
Reviewer 2 Report
Due to the high detection of Plagiarism (more than 21%) the submitted manuscript is not considered for the review.
Please resubmit the manuscript again after decreasing plagiarism to less than 10%.
Author Response
Response to Reviewer 2
Due to the high detection of Plagiarism (more than 21%) the submitted manuscript is not considered for the review. Please resubmit the manuscript again after decreasing plagiarism to less than 10%.
Answer:
We have revised the manuscript to minimize plagiarism, and we have now reduced plagiarism to 8% as shown in the enclosed PDF.
